# COVID-19 and Access to Medical Professional Careers: Does Gender Matter?

**DOI:** 10.3390/ijerph20156477

**Published:** 2023-07-31

**Authors:** Montserrat Díaz-Fernández, Mar Llorente-Marrón, Virginia Cocina-Díaz, Victor Asensi

**Affiliations:** 1Department of Quantitative Economics, University of Oviedo, 33006 Oviedo, Spain; mmarron@uniovi.es; 2Department of Sociology, University of Oviedo, 33006 Oviedo, Spain; cocinavirginia@uniovi.es; 3Department of Medicine, University of Oviedo, 33006 Oviedo, Spain; vasensi@uniovi.es

**Keywords:** access medical, profession, gender, pandemic COVID-19

## Abstract

Objective: To know to what extent home confinement resulting from the COVID-19 pandemic has affected the results of the Medical Intern Resident Program (MIR) exam and whether or not a gender gap has occurred as a consequence. Method: Econometric modeling of the final result obtained in the MIR exam and identification of the explanatory factors that determine it and its relevance, effect and meaning. Results: From the results obtained in the MIR test of the 2019, 2020 and 2021 calls, it can be seen that examinations and academic records together with demographic and calendar factors are determinants to explain the observed behavior of the final result. In relation to the gender factor, the existence of a differential fixed effect in favor of women is shown, although the interaction with the exam shows the opposite result. The nationality variable allows us to visualize a scenario of academic homogeneity. The effect of the calendar directly linked to the COVID-19 pandemic makes it possible to quantify the negative impact exerted on the final result. Conclusions: (1) The work reflects the impact of factors such as sex, nationality or the COVID-19 pandemic on access to specialized health training in Spain. (2) In contrast to previous studies, we found a significant difference in behavior between men and women, favorably linked to the female sex. However, the so-called sprint effect associated with the male sex was detected. (3) The negative effects of the COVID-19 pandemic on the final score are visualized. The existing differential with respect to the control category is quantified and the dominance of the hierarchical position of the temporal component within the set of explanatory factors is visualized.

## 1. Introduction

The process of feminization in the medical profession, traditionally male, is now a reality. The number of women in this profession has increased in most developed countries, and everything indicates that the rate of feminization will continue to rise in the coming years [1,2,3]. In the last twenty years, women have progressively joined the medical profession [1,4,5] and now represent 50% of the medical population in countries such as the United Kingdom, the Netherlands and Canada [6,7,8]. While two out of every three doctors retiring are men, two out of every three doctors entering the profession are women. In Spain, currently seven out of every ten students enrolled in medical schools are women and their presence has also been transferred to the group of resident physicians, where approximately six out of every ten residents are women [9]. 

However, the greater presence of women in the medical profession has not been reflected in a uniform distribution in professional practice according to the different medical [7] and surgical specialties [10,11,12]. Some studies suggest that this inequality is due to the different motivations, aspirations and/or preferences of men and women when choosing a career [7,13,14,15,16]. Other studies place the origin of the inequality in the different stages of academic training [13,17,18]. In this line, the existing gender gap in the academic results obtained would be the predominant cause, justified on the basis of the different behavioral patterns of men and women in competitive situations [19,20]. Generally, women show greater risk aversion [21,22,23,24] as well as higher levels of test anxiety [25]. However, each situation should be analyzed individually, since in some countries and/or areas this gender gap has been narrowing over time, and even disappearing [26,27].

In Spain, the notable growth in the number of women practicing medicine (30.41% in 1990; 52.84% in 2021) has not been reflected in terms of representation, recognition, or decision-making in a traditionally male model [21,28,29]. Although the structure of preferences in the choice of specialty could, at least in part, justify this inequality, some studies look at access to Specialized Health Training (SHT) through the MIR Program as one of the main causes of this imbalance [3]. In highly competitive environments, women are on average more risk-averse and indecisive than men, thereby obtaining worse results [23,30].

More recently, the context derived from the COVID-19 pandemic has also been the subject of analysis with reference to the learning of medical students. The effects of the periods of confinement imposed on the population by the various governments to tackle the pandemic prompted new non-face-to-face teaching tools with a clear impact on the learning process [31]. A study of 2721 students from 39 medical schools in the United Kingdom showed that more than 75% considered that online teaching did not replace the face-to-face system in which direct contact with the patient was present [32]. In the USA, through a sample obtained from six medical schools, it was found that 61.4% of students in clinical rotations felt the abrupt change in the teaching system derived from the pandemic in the development of their skills [32]. Some studies suggest that the negative effects on the learning process especially affect women [31]. Their differential impact is reflected in academic results and also in the choice of medical specialty, where the guidance usually provided by mentors during face-to-face clinical rotations is a determining factor [31].

The COVID-19 pandemic has had an abrupt impact on our society, causing changes with relevant and unforeseeable consequences. The impact of the disease and confinement, in addition to health consequences, has had consequences of an economic and social nature. The MIR exam has also been affected. This exam takes place approximately 6 months after the end of the medical degree studies. The so-called Clinical Rotation is carried out in the last year of the degree and allows students to integrate theoretical knowledge previously acquired in a real clinical practice environment.

The first signs of COVID-19 were glimpsed in January 2020, although they did not yet foreshadow the expansion that would follow. In January 2020, immediately prior to the explosion of the pandemic, the MIR exam for 2019 was held. The exam for the 2020 call was held in March 2021, in the context of a return to a new normality after two states of alarm, with the confinement of the population and the establishment of restriction measures and population control. The students who attended this call saw their clinical rotations suspended in the month of March 2020, and the preparation for the MIR exam was carried out in a virtual context. The exam for the 2021 call (January 2022) was held in a mixed scenario. The students saw face-to-face teaching activity being paralyzed in March 2020 and the clinical rotations of the 2020–2021 course interrupted. The preparation for the MIR exam was carried out in a face-to-face context.

The 2020 and 2021 calls were also marked by the COVID-19 pandemic, although in a different way. Although limitations related to the confinement of the population and social distancing rules influenced the learning patterns of students in both editions, the 2021 call was, a priori, the most affected. Applicants were affected in two academic years and, very prominently, in clinical rotations, which were mostly not carried out. All applicants for the MIR exam also had to rigorously respect the protocol established by the Ministry of Health, which has not offered alternatives to candidates who, due to being positive for coronavirus, were not able to take the test.

The implementation and official recognition in Spain of the residency system as a means of access to the title of specialist has been a very important advancement in the field of SHT and is one of the keys to the prestige and recognition currently enjoyed by professionals in the National Health System [33]. The assignment of candidates to MIR positions by specialty and hospital center is based on an orderly selection based on the results of a national test-type examination, prepared under the principles of merit and ability, and the academic scale during the degree/graduate degree in Medicine. The final grade is obtained on the basis of the academic record and the results of a national multiple-choice examination. The choice of specialty is therefore based on the principle of vertical equity by allowing the most productive candidates to have priority in the choice of the SHT program, using the ranking position as a proxy variable.

However, despite its relevance, the SHT has not received much attention in the literature, and the tests leading to its access have received even less. Bearing in mind that the grade obtained in the MIR determines the order and position of students in a list for the selection of specialties, achieving a good position in this ranking allows access to the most in-demand and/or most prestigious specialties. This article seeks to cover precisely, at least in part, the existing gap in the treatment of some of these issues. The crisis resulting from the COVID-19 pandemic has led to a change in the functioning not only of society but also of the people who make it up. To know to what extent the home confinement derived from the COVID-19 pandemic has affected the results of the MIR and whether a gender gap has been produced is consequently the objective of this work. Knowing how much and in what way the weight of the factors that explain the result derived from the confinement has been altered will allow an approximation to the future structure of the health system, insofar as the weighting of factors such as the candidate’s nationality or sex may be altered.

## 2. Materials and Methods

### 2.1. Data

The definitive list of the results of the selective tests in Medicine prepared by the Ministry of Health for the annual calls for the period 2019–2021 [34] is the source of the statistical information used. The registry provides individualized examination data, demographic information of the candidates (sex and nationality) and also calendar information. The selected period corresponds to the three exams held in the period 2020–2022 relating to the calls held in 2019, 2020 and 2021, respectively. The study was retrospective and used only data from official sources. Therefore, ethical permission from the Ministry of Health of Spain and informed consent from each individual participant were not needed. In addition, the name of each MIR examination participant was marked before data statistical analysis.

Bearing in mind that our objective was focused on identifying the explanatory factors that bring a candidate’s final score closer to access to the SHT, for the purposes of analysis, only the total number of candidates who passed the test, obtaining a score equal to or higher than the minimum mark set (35% of the arithmetic mean of the 10 best exams), were considered. Specifically, a total of 32,909 observations were considered: 12,172 observations related to the 2019 call, 10,805 for the year 2020 and 9932 for the year 2021, respectively.

### 2.2. Method

Our objective was to analyze quantitatively to what extent home confinement because of the COVID-19 pandemic affected the results of the MIR exam and whether a gender gap has been produced as a consequence. To do so, we used an econometric approach to establish a causal relationship between the result obtained and the variables that determined it. 

In the SHT job placement process, physicians choose their preferred training program based on their position in a pre-established ranking. The order of classification is a function of the average of the academic record in their studies in Medicine (10%) and of the score obtained in the MIR exam (90%). Therefore, the final result is the result of two efforts, Long-Term Effort (Baremo) and Sprint Effort (Valid). The first of these is obtained in a 6-year academic training period, while the second is the result of a test, the preparation for which by the students takes less than a year. In addition, the demographic variables *Sex*, *Nationality* and calendar, *MIR_2020* and *MIR_2021*, contribute with greater or lesser intensity to the result (Table 1). The effect of the calendar was approximated from the annual data, corresponding to the calls made in 2019, 2020 and 2021, variables *MIR_2019* and *MIR_2020*, with exams held in March 2021 (*MIR_2020 = 1*) and *MIR_2021* (*MIR_2021 = 1*) with the 2019 call for the control category. Finally, to make the sprint effect and the background effect visible according to the gender of the student, as well as the impact that the COVID-19 pandemic had on the results for each sex, interaction terms of the variable with the variables *Valid*, *Scale*, *MIR_2020* and *MIR_2021*, respectively, were included in the model.

Under the linearity assumption,
$$\begin{array}{l} Score_{i}=\beta_{0}+\beta_{1}\;Valid_{i}+\beta_{2}\;Scale_{i}+\beta_{3}\;Sex_{i}+\beta_{4}\;Nationality_{i}+\\ \;+\beta_{5}\;MIR\_2020_{i}+\beta_{6}\;MIR\_2021_{i}+\beta_{7}\;Sex_{i}\times Valid_{i}+\;\beta_{8}\;Sex_{i}\times Scale_{i}\\ \;+\beta_{9}\;Sex_{i}\times MIR\_2020_{i}+\beta_{10}\;Sex_{i}\times MIR\_2021_{i}+u_{i} \end{array}$$
will be the regression model to be estimated, where, for the *i-sima* observation, the variable that includes the final score obtained in the test, *Score*, denotes the dependent variable, and *u* the random disturbance term.

## 3. Estimation and Results 

The descriptive statistics corresponding to the variables selected from a sample of 32,909 observations are shown in Table 2. For qualitative variables, information on the number of observations that satisfy the characteristics analyzed and their relative frequency is shown. For quantitative variables, mean, median, interquartile range, minimum value, maximum value and sample standard deviation are the indicators considered.

During the selected period, participation had been decreasing both in absolute and relative terms. Of a total of 32,909 applicants who opted for an SHT position, 36.98% corresponded to the 2019 call. In the following calls, the weight was slightly lower. Between the initial and the final call, the total number of applicants decreased by 6.79 percentage points. Of the total sample, 64.12% of the applicants were women and 79.85% were Spanish nationals. The median number of valid answers and scale in the period was 119 correct answers (RIC: 103–132) and 7.49 points (RIC: 6.99–8.01), respectively. During the period analyzed, the correlation between the qualitative factor and each of the quantitative factors, *Valid* and *Scale*, presented results of the opposite sign. Results were directly proportional with the variable *Valid* (*r_Sex-Valid_* = 0.0135) and of the opposite direction to the variable Scale (*r_Sex-Scale_* = −0.0298). As a result of the high degree of correlation between the variables Sex×Valid and Sex×Scale, 0.984276, it was decided to disregard the latter in the estimation. Its inclusion in the model would generate an important multicollinearity problem captured by the Variance Inflation Factor (VIF = 454.03).

Table 3 shows the results obtained from the estimation of the model with standard errors and robust covariances, given the cross-sectional nature of the sample support used. From the analysis of the results, it can be concluded that the selected factors related to the exam (*Valid*), academic record (*Scale*), demographic (*Sex* and *Nationality*) and calendar (*MIR_2020* and *MIR_2021*) factors, as well as the interaction of the qualitative factor (*Sex*) with explanatory factors related to the exam result itself (Sex×Valid) and the calendar ($\left(Sex\times MIR_{2020}\right)$ and $\left(Sex\times MIR_{2021}\right)$), constitute relevant determinants to explain the observed behavior of the final result obtained in the test taken, *Score*, the variable to be explained. The regression is globally significant (F* = 287392.5) for a confidence level of 99.99% and the collinearity indicators do not show severe problems of linear association between the explanatory variables of the model.

The effects on the final result of the exam and academic record component were collected through the variables *Valid* and *Scale*, whose effect on the final result of the test was visualized in both cases as directly proportional and statistically significant (*Valid*: 0.7192; *p*-value < 0.0001; *Scale*: 1.1758; *p*-value < 0.0001). The effect of the applicant’s sex was approximated by the variable *Sex*, showing its effect in on model as statistically significant (*Sex*: −0.3622; *p*-value = 0.0587). The result obtained showed the differential effect on the final score between women (control category) and men. Under ceteris paribus conditions, the difference in the total score obtained between a woman and a man was 0.3622 points lower for the latter. However, the interaction of this qualitative factor with the variable that approximates the number of correct scores obtained in the exam was of the opposite sign (Sex×Valid: 0.0030; *p*-value = 0.0110). The interaction between the two factors reflected the differential effect between a woman’s and a man’s test with respect to the number of correct answers obtained. An interaction with a positive sign suggests that, under ceteris paribus conditions, the result (*Score*) for an additional valid response increased for males by 0.0030 points. The global effect of the Sex variable obtained from the contribution of both coefficients and the Redundant Variables Test (F* = 4.453125, *p*-value = 0.0116) indicates its statistical significance.

The nationality of the applicant was approximated by the variable *Nationality*, and its effect on the result was not statistically significant (*p*-value = 0.9773). The contribution of the calendar to the result was captured by the variables *MIR_2020* and *MIR_2021*, which collected observations corresponding to the test held in March 2021 and January 2022, respectively. The result obtained was positive and statistically significant, although the sign changed from one call to another (*MIR_2020*: 0.6036; *p*-value = 0.0000; *MIR_2021*: −10.6753; *p*-value = 0.0000). The result obtained showed better results for the call performed in a COVID-19 context compared to the one performed in the pre-pandemic context. Under ceteris paribus conditions, the result of the test corresponding to the 2022 call (*MIR_2021*) was 10.6753 points lower than the others. The interaction effects of the calendar factor and the *Sex* variable were collected through the variables $Sex\times MIR_{2020}$ and $Sex\times MIR_{2021}$, respectively, whose effect on the final result of the call was not statistically significant.

The standardized coefficients (Table 4) approximate the change, in typical scores, that will occur in the dependent variable for each change of one unit in the corresponding independent variable, under ceteris paribus conditions. Sorting in descending order allows us to approximate the role of the explanatory factor considered in relation to the others. The factors MIR_2021 and MIR_2020 occupied the second and fourth position, respectively, out of a total of nine variables.

## 4. Discussion

Access to the SHT in Spain is by means of an orderly selection based on the results of a multiple-choice examination and the average academic record obtained [35,36]. However, various studies suggest the existence of gender inequalities in the medical profession. Our aim in this study is to analyze the extent to which home confinement due to the COVID-19 pandemic affected the results of the MIR and whether or not a gender gap was produced as a consequence.

Based on the results obtained in the MIR exams of the 2019, 2020 and 2021 calls, an empirical exercise was carried out that econometrically modeled the final result of the test. The econometric approach allowed us to perform a causal analysis and identify and classify the explanatory factors that contributed to explaining the final score and its relevance, effect and meaning [37]. Although the final result is obtained as the sum of the final exam score and academic record, it is also a consequence of the interaction of the examinees with the real environment, an aspect that allows the econometric approach to be integrated. By means of econometric techniques, in addition to the quantitative variables that make up the deterministic equation, it was possible to incorporate into the analysis the effect generated by qualitative factors, such as gender, nationality or calendar.

In the exercise carried out, the Valid and Scale variables that made up the deterministic equation that defined the final score obtained were identified as relevant factors in the MIR result. Some studies identify these variables with the so-called sprint effort and background effort, respectively [3,38]. The results obtained showed that the effect of an additional valid response or one more point on the academic record, under ceteris paribus conditions, would generate an increase in the final score. Although the effect was increasing in both cases, for the Valid variable, it was less than proportional. In standardized terms, however, the weight of the exam outweighs the academic record. The sprint effort relative to background effort is dominant in the results obtained in line with studies that highlight the different risk exposure of men and women [23,30].

Our results visualized the effect on the final score attributed to the sex variable. The sign and statistical significance of this variable, under ceteris paribus conditions, showed the existence of a differential fixed effect favorably linked to female sex. Nevertheless, some studies suggest that being male improves the final result because historically the academic results of women have been lower [17,18,36,39]. The interaction term between the Sex and Valid variable visualizes the differential effect corresponding to the number of correct answers between the final score of a woman and a man. A positive interaction suggests that the final score of a man increases relative to that of a woman with respect to the variable Valid, while a negative interaction would suggest the opposite. The result of the MIR exam, a unique test in a highly competitive environment, is the result of a relatively short but very intense preparation period, defined as a Sprint Effort. Our result reaffirms those obtained in studies that indicate that increasing the level of competition improves male performance, while women, with greater risk aversion, do not react in the same way. This could be explained by differences in risk and confidence between the sexes. Women show greater aversion to risk in competitive environments, while men increase their participation and performance [30,40,41,42]. However, in less competitive or stressful scenarios, such as, for example, exams taken during the course, women perform better than men [43,44]. The sex factor is configured as a determining factor whose effect on the final result is divided into two components: the existence of a fixed differential factor favorably linked to the female sex, and its interaction with the total number of correct answers (sprint effect) favorable to the male sex. 

The effect of the variable Nationality on the final result was not statistically significant. The Ministry of Health requires the homologation of the degree in Medicine as a requirement to be able to take the MIR exam and access an SHT position, a necessary condition for professional practice in Spain. Being Spanish or not does not favor or reduce the final grade obtained. The result is framed within a scenario of harmonization and homogeneity of academic training in the international context [45,46]. 

The COVID-19 pandemic has had a profound impact on our society, causing changes with relevant and sometimes unforeseeable consequences. The impact of disease and confinement, in addition to health consequences, has had consequences of an economic and social nature. The incorporation into the model of factors that approximate the timing of the test makes it possible to know how the performance of the test on one date or another has affected the final result. Since the first patient in Spain was diagnosed with COVID-19 coronavirus, in January 2020, the evolution of the pandemic has gone through two states of alarm, with confinement of the population and the adoption of prevention, containment and coordination measures. The selected period of analysis began in a pre-pandemic setting [47,48]. The review of the 2019 call was conducted on 25 January 2020, at the time of the pandemic explosion and peak, although the adoption of measures and containment of the population did not occur until eight weeks later, 14 March. Linked to the 2020 call, the examination held in March 2021 corresponds, after an abrupt experience, with the return to a scenario of new normality in which different scenarios are glimpsed. In that call, two states of alarm were experienced; the first with general confinement of the population (14 March 2020/21 June 2021), and the second with different measures of protection and containment of population movements (25 October 2020/May 2021). The pandemic containment measures contemplated, in addition to other issues, the replacement of face-to-face teaching activity at all educational levels. For the 2020 call, these measures corresponded to the suspension of clinical rotations from March 2020 and the virtual preparation of the MIR exam. Finally, for the 2021 call, the exam was returned to the usual date and was held in January 2022. This call included the second state of alarm, as well as the de-escalation in the measures implemented to contain the population, with modifications and restrictions in day-to-day activities. Applicants to this call saw their penultimate academic year interrupted in March 2020, and clinical rotations to be performed in the 2020–2021 academic year were interrupted. Finally, the uncertainty with which the horizon looms is the common factor of the accumulated effects on the behavioral patterns of society in general, and of the MIR exam candidates in particular.

The calendar effect was shown in the model to be statistically significant. Some studies suggest a negative impact on academic performance, among the effects of the pandemic [49]. The sign and statistical significance of the variables MIR_2020 and MIR_2021 in ceteris paribus conditions showed the existence of a positive differential in the 2020 call and a negative one in the following call, with respect to the 2019 call. As a consequence of the pandemic, students in general and MIR exam candidates in particular saw their learning process conditioned and were forced to replace traditional teaching tools and methods with virtual or blended learning environments in a very short period of time [50]. Different studies have suggested that the state of mind of the students during the pandemic did not favor study and preparation for the tests [51,52,53]. Our result allowed us to quantify the effect of these changes by visualizing a differential in the lower final score for the candidates taking the 2021 exam compared to the control. They were the most affected in the process as a consequence of the pandemic. Face-to-face teaching and clinical rotations were suspended from March 2020 to June 2021.

The analysis made it possible to visualize differences in the results linked to the time reference. It also showed the determinant role the time reference plays in the model, by occupying the second position in the ranking derived from the estimation of the standardized coefficients between the quantitative variables *Valid* and *Scale*. The interaction term between the *Sex* variable and the calendar did not show statistically significant results. No COVID effect by sex was detected. COVID-19 did not show an additional effect on the influence of the *Sex* factor on the final result of the MIR.

## 5. Conclusions 

This work has highlighted the impact of factors such as gender, nationality or the COVID-19 pandemic on access to the SHT in Spain. In contrast to previous studies, we found a significant difference in behavior between men and women in terms of the existence of a fixed differential favorably linked to female sex. However, the results obtained reaffirmed that, in highly competitive environments, the so-called sprint effort improves the performance of men in relation to women. Our study also allowed us to visualize the negative effects of the COVID-19 pandemic on the final score, quantifying, on the one hand, the existing differential with respect to the control category, and on the other hand, visualizing the hierarchical position of the temporal component within the set of explanatory factors. The results showed no significant differences in access linked to the nationality of the applicant. 

Identifying, measuring and classifying the explanatory factors that determine the final score and position of MIR place assignment is essential to identify the stage that initiates professional practice in the health field. Analyzing the existence of gender inequalities in this process allows, on the one hand, the identification of whether the MIR system contributes to perpetuating the underrepresentation of women in the most demanded specialties, and on the other hand, to the development of public policies that ensure more egalitarian procedures. The effects of the COVID-19 pandemic on the academic results of medical studies highlight the importance of social interaction and social learning in medical education.

Future research should explore the specialty assignment process, since its characteristics could favor one group over another. In addition, we believe that it would be interesting to include in the analysis those candidates who have not passed the test, in addition to carrying out an analysis by medical specialties.

## Figures and Tables

**Table 1 ijerph-20-06477-t001:** Selected variables used in the study.

Indicator	Variable	Definition	Expected Sign
Test	*Valid*	Quantitative variable indicating the total number of correct answers obtained by each candidate	The expected sign of this variable is positive
	*Scale*	Quantitative variable obtained by multiplying the evaluation of each candidate’s file by 10 and dividing the product by the average of the ten best exams of the corresponding exercise	
Demographic factors	*Sex*	Dichotomous variable that takes the value 0 if the applicant is female, 1 otherwise	A priori, the expected sign of this variable can be + or –
	*Nationality*	Dichotomous variable that takes the value 0 if the applicant is of Spanish nationality, 1 otherwise	
Calendar	*MIR_2020*	Dichotomous variable that takes the value 1 if the sample observation corresponds to the 2020 call, otherwise 0	A priori, the expected sign of this variable can be + or −
	*MIR_2021*	Dichotomous variable that takes the value 1 if the sample observation corresponds to the call for the year 2021, 0 otherwise	

**Table 2 ijerph-20-06477-t002:** Descriptive statistics of the sample, period 2019–2021 (*n* = 32,909).

Indicator		Period 2019–2021 (*n* = 32,909)					
Demographic factors	**Qualitative** **Variables**	**Number of Cases**			**Frequency %**		
	*Sex*	Woman		21,104	64.12		
		Men		11,805	35.88		
	*Nationality*	Spanish		26,278	79.85		
		Foreign		6631	20.15		
Calendar	*MIR call*	*MIR_2019*		12,172	36.98		
		*MIR_2020*		10,805	32.83		
		*MIR_2021*		9932	30.19		
Test	**Quantitative Variables**	**Mean**	**Median**	**RIC**	**Minimum**	**Maximum**	**Std Deviation**
	*Valid*	118.05	119.00	103–132	65.00	182.00	19.24
	*Scale*	7.41	7.49	6.99–8.01	5.00	10.00	0.92

RIC: interquartile range. Source: own elaboration based on Ministry of Health data (2022).

**Table 3 ijerph-20-06477-t003:** Ordinary Least Squares Estimation. Final score, period 2019-2021.

Dependent Variable: *Score*				
Variable	Coefficient	Std. Error	t-Statistic	Prob.
C	67.38699	0.013463	5005.424	0.0000
*Valid*	0.719289	0.001214	592.6089	0.0000
*Scale*	1.175895	0.015254	77.08963	0.0000
*Sex*	−0.362214	0.191619	−1.890279	0.0587
*Nationality*	−0.000772	0.027128	−0.028448	0.9773
*MIR_2020*	0.603623	0.020200	29.88237	0.0000
*MIR_2021*	−10.67533	0.032173	−331.8055	0.0000
Sex×Valid	0.003035	0.001643	1.847576	0.0110
$Sex\times MIR_{2020}$	−0.000831	0.042103	−0.019736	0.9843
$Sex\times MIR_{2021}$	−0.064417	0.064889	−0.992732	0.3208
R-squared	0.985387	Mean dependent var.		64.39472
Adjusted R-squared	0.985383	S.D. dependent var.		13.61364
S.E. of regression	1.645913	Akaike info criterion		3.834772
Sum squared resid.	89,045.83	Schwarz criterion		3.837327
Log likelihood	−63,033.65	Hannan-Quinn criter.		3.835588
F-statistic	246,273.3	Durbin–Watson stat.		1.996789
Prob(F-statistic)	0.000000	Wald F-statistic		124,176.6
Prob(Wald F-statistic)	0.000000			

Method: Ordinary Least Squares. Huber–White–Hinkley (HC1) heteroskedasticity consistent standard error and covariance. Sample size: MIR_2019 (12,172); MIR_2020 (10,805); MIR_2021 (9932). Observations included: 32,880. Source: own elaboration based on Ministry of Health data (2022).

**Table 4 ijerph-20-06477-t004:** Standardized coefficients.

Variable	Coefficient	Standardized Coefficient	Range
C	67.38699	NA	
*Valid*	0.719289	1.016319	1
*Scale*	1.175895	0.079978	3
*Sex*	−0.362214	−0.012756	6
*Nationality*	−0.000772	−2.27 × 10^−5^	8
*MIR_2020*	0.603623	0.020827	4
*MIR_2021*	−10.67533	−0.359770	2
Sex×Valid	0.003035	0.012930	5
$Sex\times MIR_{2020}$	−0.000831	−1.68 × 10^−5^	9
$Sex\times MIR_{2021}$	−0.064417	−0.001239	7

Source: Own elaboration based on Ministry of Health data (2022).

## Data Availability

Ministerio de Sanidad. *Profesionales—Formación Especializada*. Available online: https://www.sanidad.gob.es/areas/profesionesSanitarias/formacionEspecializada/home.htm (accessed on 1 March 2022).

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
