# Peer review of "COVID-19 and Access to Medical Professional Careers: Does Gender Matter?"

_ijerph, 2023, doi:10.3390/ijerph20156477_

Round 1
Reviewer 1 Report
Thank you for giving me to opportunity to review your manuscript, it was indeed a very interesting read.
Introduction: The topic is very interesting, especially the exploration of the gender differences during the COVID-19 pandemic in medical education.
Please provide the whole explanation on abbreviations at least one time at the first instance they are met in the manuscript (e.g. ESF).
This is an EU study. please consider the following very current EU projects to strengthen this section with relevant research:
Chatzi, A. V., & Murphy, C. (2022). Investigation on gender and area of study stereotypes among Irish third level students. International Journal of Educational Research Open, 3, 100171. doi:https://doi.org/10.1016/j.ijedro.2022.100171
Clavel, J. G., & Flannery, D. (2023). Single‐sex schooling, gender and educational performance: Evidence using PISA data. British Educational Research Journal, 49(2), 248-265.
Material and Method, Data: Please provide more information on the data used (how they are collected, exclusion inclusion criteria and ethics approval).
Discussion: Please explain terms such as sprint effort (line 257, 262, 338) and background effort (line 258, 263).
Lines 273, 274: “Our result reaffirms those obtained in studies indicating that, in highly competitive environments, the so-called sprint effort improves male performance 21,28.” Please rewrite for clarity.
Is the effect of gender in exam results statistically significant? Please give it more clearly in the text.
Conclusions: Line 342: “visualizing the hierarchical position of the temporal component within the set of explanatory factors.” Please explain.
Please include, from the findings in the project, more specific recommendations for further research.
Very good quality of English language. A couple of sentences that need to be re-written for clarity have already been given in previous field to authors.
Author Response
Dear Reviewer, we appreciate your time and effort in reviewing our paper and greatly value your constructive comments. We found them insightful and useful. We respond to your remarks below.
Three files are attached with the answers and modifications made.
Thank you!

Reviewer 2 Report
The work is well-written and talks about a very interesting topic.
However, I would like to point out some points that need to be corrected:
1) I ask the authors to specify the demographic information (line 128).
2) I ask the authors to specify if significant differences exist between those who have been considered and those who have not been considered for the analysis (lines 132-135).
3) I invite the authors to use the same acronyms throughout the text (specialized health training = FSE in line 57 and ESF in line 111).
4) I point out an error in line 258 and line 275 about references.
5) The authors should also add a section regarding the limits of the study (I suggest, for example, the low number of demographic variables in results section and the little weight that can be given to the nationality variable (are we talking about foreigners who arrived in Spain to do the MIR or foreigners who have been present in the country for some time?).
6) I suggest adding a quote to line 292 to support the sentence.
7) The term "convocatoria" in table 2 is not an English term
Author Response

(The authors gave the same response as above.)
